# Integrated Perception with Recurrent Multi-Task Neural Networks

**Hakan Bilen**  **Andrea Vedaldi**
Visual Geometry Group, University of Oxford
{hbilen,vedaldi}@robots.ox.ac.uk

## Abstract

Modern discriminative predictors have been shown to match natural intelligences in specific perceptual tasks in image classification, object and part detection, boundary extraction, etc. However, a major advantage that natural intelligences still have is that they work well for *all* perceptual problems together, solving them efficiently and coherently in an *integrated manner*. In order to capture some of these advantages in machine perception, we ask two questions: whether deep neural networks can learn universal image representations, useful not only for a single task but for all of them, and how the solutions to the different tasks can be integrated in this framework. We answer by proposing a new architecture, which we call *multinet*, in which not only deep image features are shared between tasks, but where tasks can interact in a recurrent manner by encoding the results of their analysis in a common shared representation of the data. In this manner, we show that the performance of individual tasks in standard benchmarks can be improved first by sharing features between them and then, more significantly, by integrating their solutions in the common representation.

## 1 Introduction

Natural perception can extract complete interpretations of sensory data in a coherent and efficient manner. By contrast, machine perception remains a collection of disjoint algorithms, each solving specific information extraction sub-problems. Recent advances such as modern convolutional neural networks have dramatically improved the performance of machines in individual perceptual tasks, but it remains unclear how these could be *integrated* in the same seamless way as natural perception does.

In this paper, we consider the problem of learning data representations for integrated perception. The first question we ask is whether it is possible to learn *universal data representations* that can be used to solve all sub-problems of interest. In computer vision, fine-tuning or retraining has been show to be an effective method to transfer deep convolutional networks between different tasks [9, 29]. Here we show that, in fact, it is possible to learn a single, shared representation that performs well on several sub-problems *simultaneously*, often as well or even better than specialised ones.

A second question, complementary to the one of feature sharing, is how different perceptual subtasks should be combined. Since each subtask extracts a partial interpretation of the data, the problem is to form a coherent picture of the data as a whole. We consider an incremental interpretation scenario, where subtasks collaborate in parallel or sequentially in order to gradually enrich a shared interpretation of the data, each contributing its own "dimension" to it. Informally, many computer vision systems operate in this stratified manner, with different modules running in parallel or in sequence (e.g. object detection followed by instance segmentation). The question is how this can be done end-to-end and systematically.

In this paper, we develop an architecture, *multinet* (fig. 1), that provides an answer to such questions. Multinet builds on the idea of a shared representation, called an integration space, which reflects both

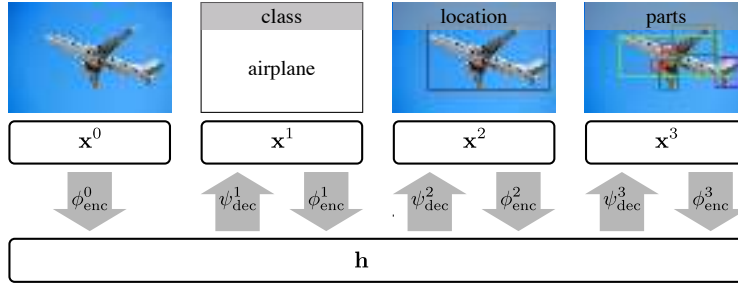

Figure 1: **Multinet.** We propose a modular multi-task architecture in which several perceptual tasks are integrated in a synergistic manner. The subnetwork $\phi_{enc}^0$ encodes the data $\mathbf{x}^0$ (an image in the example) producing a representation $\mathbf{h}$ shared between $K$ different tasks. Each task estimates one of $K$ different labels $\mathbf{x}^\alpha$ (object class, location, and parts in the example) using $K$ decoder functions $\psi_{dec}^\alpha$. Each task contributes back to the shared representation by means of a corresponding encoder function $\phi_{enc}^\alpha$. The loop is closed in a recurrent configuration by means of suitable integrator functions (not shown here to avoid cluttering the diagram).

the statistics extracted from the data as well as the result of the analysis carried by the individual subtasks. As a loose metaphor, one can think the integration space as a "canvas" which is progressively updated with the information obtained by solving sub-problems. The representation distills this information and makes it available for further task resolution, in a recurrent configuration.

Multinet has several advantages. First, by learning the latent integration space automatically, synergies between tasks can be discovered automatically. Second, tasks are treated in a symmetric manner, by associating to each of them encoder, decoder, and integrator functions, making the system modular and easily extensible to new tasks. Third, the architecture supports incremental understanding because tasks contribute back to the latent representation, making their output available to other tasks for further processing. Finally, while multinet is applied here to a image understanding setting, the architecture is very general and could be applied to numerous other domains as well.

The new architecture is described in detail in sect. 2 and an instance specialized for computer vision applications is given in sect. 3. The empirical evaluation in sect. 4 demonstrates the benefits of the approach, including that sharing features between different tasks is not only economical, but also sometimes better for accuracy, and that integrating the outputs of different tasks in the shared representation yields further accuracy improvements. Sect. 5 summarizes our findings.

## 1.1 Related work

**Multiple task learning (MTL):** Multitask learning [5, 25, 1] methods have been studied over two decades by the machine learning community. The methods are based on the key idea that the tasks share a common low-dimensional representation which is jointly learnt with the task specific parameters. While MLT trains many tasks in parallel, Mitchell and Thrun [18] propose a sequential transfer method called Explanation-Based Neural Nets (EBNN) which exploits previously learnt domain knowledge to initialise or constraint the parameters of the current task. Breiman and Freidman [3] devise a hybrid method that first learns separate models and then improves their generalisation by exploiting the correlation between the predictions.

**Multi-task learning in computer vision:** MTL has been shown to improve results in many computer vision problems. Typically, researchers incorporate auxiliary tasks into their target tasks, jointly train them in parallel and achieve performance gains in object tracking [30], object detection [11], facial landmark detection [31]. Differently, Dai *et al.* [8] propose multi-task network cascades in which convolutional layer parameters are shared between three tasks and the tasks are predicted sequentially. Unlike [8], our method can train multiple tasks in parallel and does not require a specification of task execution.

**Recurrent networks:** Our work is also related to recurrent neural networks (RNN) [22] which has been successfully used in language modelling [17], speech recognition [13], hand-written recognition [12], semantic image segmentation [20] and human pose estimation [2]. Related to our work, Carreira *et al.* [4] propose an iterative segmentation model that progressively updates an initial

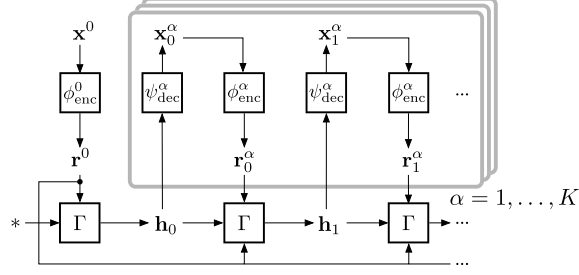

Figure 2: **Multinet recurrent architecture**. The components in the rounded box are repeated $K$ times, one for each task $\alpha = 1, \ldots, K$.

solution by feeding back error signal. Najibi *et al.* [19] propose an efficient grid based object detector that iteratively refine the predicted object coordinates by minimising the training error. While these methods [4, 19] are also based on an iterative solution correcting mechanism, our main goal is to improve generalisation performance for multiple tasks by sharing the previous predictions across them and learning output correlations.

## 2 Method

In this section, we first introduce the multinet architecture for integrated multi-task prediction (sect. 2.1) and then we discuss ordinary multi-task prediction as a special case of multinet (sect. 2.2).

### 2.1 Multinet: integrated multiple-task prediction

We propose a recurrent neural network architecture (fig. 1 and 2) that can address simultaneously multiple data labelling tasks. For symmetry, we drop the usual distinction between input and output spaces and consider instead $K$ *label spaces* $\mathcal{X}^\alpha, \alpha = 0, 1, \ldots, K$. A label in the $\alpha$-th space is denoted by the symbol $\mathbf{x}^\alpha \in \mathcal{X}^\alpha$. In the following, $\alpha = 0$ is used for the input (e.g. an image) of the network and is not inferred, whereas $\mathbf{x}^1, \ldots, \mathbf{x}^K$ are labels estimated by the neural network (e.g. an object class, location, and parts). One reason why it is useful to keep the notation symmetric is because it is possible to ground any label $\mathbf{x}^\alpha$ and treat it as an input instead.

Each task $\alpha$ is associated to a corresponding *encoder function* $\phi_{\text{enc}}^\alpha$, which maps the label $\mathbf{x}^\alpha$ to a vectorial representation $\mathbf{r}^\alpha \in \mathcal{R}^\alpha$ given by

$$\mathbf{r}^\alpha = \phi_{\text{enc}}^\alpha(\mathbf{x}^\alpha). \tag{1}$$

Each task has also a *decoder function* $\psi_{\text{dec}}^\alpha$ going in the other direction, from a common representation space $\mathbf{h} \in \mathcal{H}$ to the label $\mathbf{x}^\alpha$:

$$\mathbf{x}^\alpha = \psi_{\text{dec}}^\alpha(\mathbf{h}). \tag{2}$$

The information $\mathbf{r}^0, \mathbf{r}^1, \ldots, \mathbf{r}^\alpha$ extracted from the data and the different tasks by the encoders is *integrated* in the shared representation $\mathbf{h}$ by using an integrator function $\Gamma$. Since this update operation is incremental, we associate to it an iteration number $t = 0, 1, 2, \ldots$. By doing so, the update equation can be written as

$$\mathbf{h}_{t+1} = \Gamma(\mathbf{h}_t, \mathbf{r}^0, \mathbf{r}_t^1, \ldots, \mathbf{r}_t^K). \tag{3}$$

Note that, in the equation above, $\mathbf{r}^0$ is constant as the corresponding variable $\mathbf{x}^0$ is the input of the network, which is grounded and not updated.

Overall, a task $\alpha$ is specified by the triplet $\mathcal{T}^\alpha = (\mathcal{X}^\alpha, \phi_{\text{enc}}^\alpha, \psi_{\text{dec}}^\alpha)$ and by its contribution to the update rule (3). Full task modularity can be achieved by decomposing the integrator function as a sequence of task-specific updates $\mathbf{h}_{t+1} = \Gamma^K(\cdot, \mathbf{r}_t^K) \circ \cdots \circ \Gamma^1(\mathbf{h}_t, \mathbf{r}_t^1)$, such that each task is a quadruplet $(\mathcal{X}^\alpha, \phi_{\text{enc}}^\alpha, \psi_{\text{dec}}^\alpha, \Gamma^\alpha)$, but this option is not investigated further here.

Given tasks $\mathcal{T}^\alpha, \alpha = 1, \ldots, K$, several variants of the recurrent architecture are possible. A natural one is to process tasks sequentially, but this has the added complication of having to choose a particular order and may in any case be suboptimal; instead, we propose to update all the task at each recurrent iteration, as follows:

$t = 0$ **Ordinary multi-task prediction.** At the first iteration, the measurement $\mathbf{x}^0$ is acquired and the shared representation $\mathbf{h}$ is initialized as $\mathbf{h}_0 = \phi_{\text{enc}}^0(\mathbf{x}^0) = \Gamma(*, \mathbf{r}^0, *, \ldots, *)$. The symbol $*$ denotes the initial value of a variable (often zero in practice). Given $\mathbf{h}^0$, the output $\mathbf{x}_0^\alpha = \psi_{\text{dec}}^\alpha(\mathbf{h}_0) = (\psi_{\text{dec}}^\alpha \circ \phi_{\text{enc}}^0)(\mathbf{x}^0)$ for each task is computed. This step corresponds to ordinary multi-task prediction, as discussed later (sect. 2.2).

$t > 0$ **Iterative updates.** Each task $\alpha = 1, \ldots, K$ is re-encoded using equations $\mathbf{r}_t^\alpha = \phi_{\text{enc}}^\alpha(\mathbf{x}_t^\alpha)$, the shared representation is updated using $\mathbf{h}_{t+1} = \Gamma(\mathbf{h}_t, \mathbf{r}^0, \mathbf{r}_t^1, \ldots, \mathbf{r}_t^K)$, and the labels are predicted again using $\mathbf{x}_{t+1}^\alpha = \psi_{\text{dec}}^\alpha(\mathbf{h}_{t+1})$.

The idea of feeding back the network output for further processing exists in several existing recurrent architectures [16, 24]; however, in these cases it is used to process sequential data, passing back the output obtained from the last process element in the sequence; here, instead, the feedback is used to integrate different and complementary labelling tasks. Our model is also reminiscent of encoder/decoder architectures [15, 21, 28]; however, in our case the encoder and decoder functions are associated to the output labels rather than to the input data.

## 2.2 Ordinary multi-task learning

Ordinarily, multiple-task learning [5, 25, 1] is based on sharing features or parameters between different tasks. Multinet reduces to ordinary multi-task learning when there is no recurrence. At the first iteration $t = 0$, in fact, multinet simply evaluates $K$ predictor functions $\psi_{\text{dec}}^1 \circ \phi_{\text{enc}}^0, \ldots, \psi_{\text{dec}}^K \circ \phi_{\text{enc}}^0$, one for each task, which share the common subnetwork $\phi_{\text{enc}}^0$.

While multi-task learning from representation sharing is conceptually simple, it is practically important because it allows learning a *universal representation function* $\phi_{\text{enc}}^0$ which works well for all tasks simultaneously. The possibility of learning such a polyvalent representation, which can only be verified empirically, is a non-trivial and useful fact. In particular, in our experiments in image understanding (sect. 4), we will see that, for certain image analysis tasks, it is not only possible and efficient to learn such a shared representation, but that in some cases feature sharing can even improve the performance in the individual sub-problems.

## 3 A multinet for classification, localization, and part detection

In this section we instantiate multinet for three complementary tasks in computer vision: object classification, object detection, and part detection. The main advantage of multinet compared to ordinary multi-task prediction is that, while sharing parameters across related tasks may improve generalization [5], it is not enough to capture correlations in the task input spaces. For example, in our computer vision application ordinary multi-task prediction would not be able to ensure that the detected parts are contained within a detected object. Multinet can instead capture interactions between the different labels and potentially learn to enforce such constraints. The latter is done in a soft and distributed manner, by integrating back the output of the individual tasks in the shared representation.

Next, we discuss in some detail the specific architecture components used in our application. As a starting point we consider a standard CNN for image classification. While more powerful networks exist, we choose here a good performing model which is at the same time reasonably efficient to train and evaluate, namely the VGG-M-1024 network of [6]. This model is pre-trained for image classification from the ImageNet ILSVRC 2012 data [23] and was extended in [11] to object detection; here we follow such blueprints, and in particular the Fast R-CNN method of [11], to design the subnetworks for the three tasks. These components are described in some detail below, first focusing on the components corresponding to ordinary multi-task prediction, and then moving to the ones used for multiple task integration.

**Ordinary multiple-task components.** The first several layers of the VGG-M network can be grouped in five convolutional sections, each comprising linear convolution, a non-linear activation function and, in some cases, max pooling and normalization. These are followed by three fully-connected sections, which are the same as the convolutional ones, but with filter support of the same size as the corresponding input. The last layer is softmax and computes a posterior probability vector over the 1,000 ImageNet ILSVRC classes.

VGG-M is adapted for the different tasks as follows. For clarity, we use symbolic names for the tasks rather than numeric indexes, and consider $\alpha \in \{\text{img}, \text{cls}, \text{det}, \text{part}\}$ instead of $\alpha \in \{0, 1, 2, 3\}$. The five convolutional sections of VGG-M are used as the image encoder $\phi_{\text{enc}}^{\text{img}}$ and hence compute the initial value $\mathbf{h}_0$ of the shared representation. Cutting VGG-M at the level of the last convolutional layer is motivated by the fact that the fully-connected layers remove or at least dramatically blur spatial information, whereas we would like to preserve it for object and part localization. Hence, the shared representation is a tensor $\mathbf{h} \in \mathbb{R}^{H \times W \times C}$, where $H \times W$ are the spatial dimensions and $C$ is the number of feature channels as determined by the VGG-M configuration (see sect. 4).

Next, $\phi_{\text{enc}}^{\text{img}}$ is branched off in three directions, choosing a decoder $\psi_{\text{dec}}^{\alpha}$ for each task: image classification ($\alpha = \text{cls}$), object detection ($\alpha = \text{det}$), and part detection ($\alpha = \text{part}$). For the *image classification* branch, we choose $\phi_{\text{enc}}^{\alpha}$ as the rest of the original VGG-M network for image classification. In other words, the decoder function $\psi_{\text{dec}}^{\text{cls}}$ for the image-level labels is initialized to be the same as the fully-connected layers of the original VGG-M, such that $\phi_{\text{enc}}^{\text{VGG-M}} = \psi_{\text{dec}}^{\text{cls}} \circ \phi_{\text{enc}}^{\text{img}}$. There are however two differences. The first is the last fully-connected layer is reshaped and reinitialized randomly to predict a different number $C$ of possible objects instead of the 1,000 ImageNet classes. The second difference is that the final output is a vector of binary probabilities obtained using sigmoid instead of a softmax.

The object and part detection decoders are instead based on the Fast R-CNN architecture [11], and classify individual image regions as belonging to one of the object classes (part types) or background. To do so, the Selective Search Windows (SSW) method [26] is used to generate a shortlist of $M$ region (bounding box) proposals $\mathcal{B}(\mathbf{x}^{\text{img}}) = \{\mathbf{b}_1, \ldots, \mathbf{b}_M\}$ from image $\mathbf{x}^{\text{img}}$; this set is inputted to the *spatial pyramid pooling (SPP) layer* [14, 11] $\psi_{\text{dec}}^{\text{SPP}}(\mathbf{h}, \mathcal{B}(\mathbf{x}^{\text{img}}))$, which extracts subsets of the feature map $\mathbf{h}$ in correspondence of each region using max pooling. The object detection decoder (and similarly for the part detector) is then given by $\psi_{\text{dec}}^{\text{det}}(\mathbf{h}) = \underline{\psi_{\text{dec}}}^{\text{det}}(\psi_{\text{dec}}^{\text{SPP}}(\mathbf{h}, \mathcal{B}(\mathbf{x}^{\text{img}})))$ where $\underline{\psi_{\text{dec}}}^{\text{det}}$ contains fully connected layers initialized in the same manner as the classification decoder above (hence, before training one also has $\phi_{\text{enc}}^{\text{VGG-M}} = \underline{\psi_{\text{dec}}}^{\text{det}} \circ \phi_{\text{enc}}^{\text{img}}$). The exception is once more the last layer, reshaped and reinitialized as needed, whereas softmax is still used as regions can have only one class.

So far, we have described the image encored $\phi_{\text{enc}}^{\text{img}}$ and the decoder branches $\psi_{\text{dec}}^{\text{cls}}$, $\psi_{\text{dec}}^{\text{det}}$ and $\psi_{\text{dec}}^{\text{part}}$ for the three tasks. Such components are sufficient for ordinary multi-task learning, corresponding to the initial multinet iteration. Next, we specify the components that allow to iterate multinet several times.

**Recurrent components: integrating multiple tasks.** For task integration, we need to construct the encoder functions $\phi_{\text{enc}}^{\text{cls}}$, $\phi_{\text{enc}}^{\text{det}}$ and $\phi_{\text{enc}}^{\text{part}}$ for each task as well as the integrator function $\Gamma$. While several constructions are possible, here we experiment with simple ones.

In order to encode the image label $\mathbf{x}^{\text{cls}}$, the encoder $\mathbf{r}^{\text{cls}} = \phi_{\text{enc}}^{\text{cls}}(\mathbf{x}^{\text{cls}})$ takes the vector of $C^{\text{cls}}$ binary probabilities $\mathbf{x}^{\text{cls}} \in \mathbb{R}^{C^{\text{cls}}}$, one for each of the $C^{\text{cls}}$ possible object classes, and broadcasts the corresponding values to all $H \times W$ spatial locations $(u, v)$ in $\mathbf{h}$. Formally $\mathbf{r}^{\text{cls}} \in \mathbb{R}^{H \times W \times C^{\text{cls}}}$ and

$$\forall u, v, c: \quad r_{uvc}^{\text{cls}} = x_c^{\text{cls}}.$$

Encoding the object detection label $\mathbf{x}^{\text{det}}$ is similar, but reflects the geometric information captured by such labels. In particular, each bounding box $\mathbf{b}_m$ of the $M$ extracted by SSW is associated to a vector of $C^{\text{cls}} + 1$ probabilities (one for each object class plus one more for background) $\mathbf{x}_m^{\text{det}} \in \mathbb{R}^{C^{\text{cls}}+1}$. This is decoded in a heat map $\mathbf{r}^{\text{cls}} \in \mathbb{R}^{H \times W \times (C^{\text{cls}}+1)}$ by max pooling across boxes:

$$\forall u, v, c: \quad r_{uvc}^{\text{cls}} = \max \left\{ x_{mc}^{\text{det}}, \forall m: (u, v) \in \mathbf{b}_m \right\} \cup \{0\}.$$

The part label $\mathbf{x}^{\text{part}}$ is encoded in an entirely analogous manner.

Lastly, we need to construct the integrator function $\Gamma$. We experiment with two simple designs. The first one simply *stacks* evidence from the different sources: $\mathbf{h} = \text{stack}(\mathbf{r}^{\text{img}}, \mathbf{r}^{\text{cls}}, \mathbf{r}^{\text{det}}, \mathbf{r}^{\text{part}})$. Then the update equation is given by

$$\mathbf{h}_t = \Gamma(\mathbf{h}_{t-1}, \mathbf{r}^{\text{img}}, \mathbf{r}_t^{\text{cls}}, \mathbf{r}_t^{\text{det}}, \mathbf{r}_t^{\text{part}}) = \text{stack}(\mathbf{r}^{\text{img}}, \mathbf{r}_t^{\text{cls}}, \mathbf{r}_t^{\text{det}}, \mathbf{r}_t^{\text{part}}). \tag{4}$$

Note that this formulation requires modifying the first fully-connected layers of each decoder $\hat{\psi}_{\text{dec}}^{\text{cls}}$, $\hat{\psi}_{\text{dec}}^{\text{det}}$ and $\hat{\psi}_{\text{dec}}^{\text{part}}$ as the shared representation $\mathbf{h}$ has now $C + 2C^{\text{cls}} + C^{\text{part}} + 2$ channels instead of just $C$

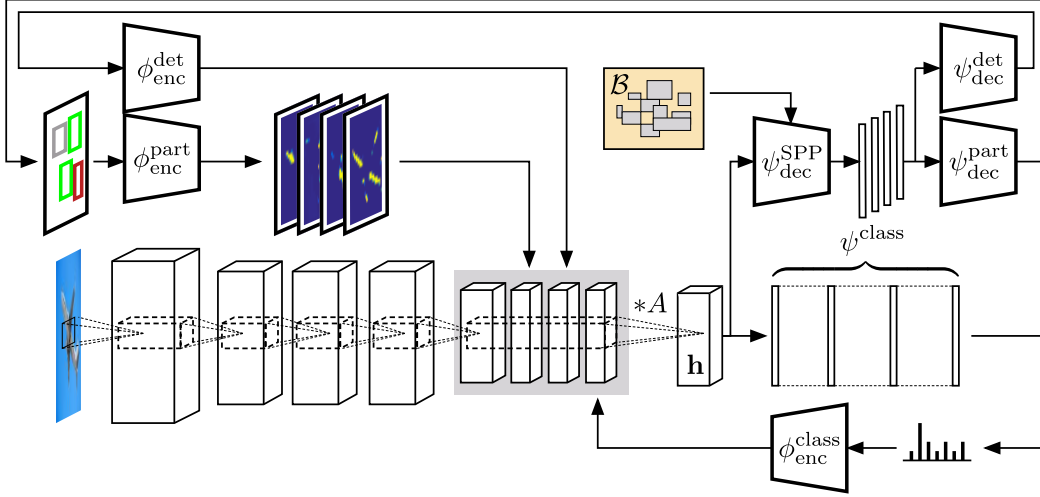

Figure 3: Illustration of the multinet instantiation tackling three computer vision problem: image classification, object detection, and part detection.

as for the original VGG-M architecture. This is done by initializing randomly additional dimensions in the linear maps.

We also experiment with a second update equation

$$\mathbf{h}_t = \Gamma(\mathbf{h}_{t-1}, \mathbf{r}^{\text{img}}, \mathbf{r}_t^{\text{cls}}, \mathbf{r}_t^{\text{det}}, \mathbf{r}_t^{\text{part}}) = \text{ReLU}(A * \text{stack}(\mathbf{h}_{t-1}, \mathbf{r}^{\text{cls}}, \mathbf{r}_t^{\text{cls}}, \mathbf{r}_t^{\text{det}}, \mathbf{r}_t^{\text{part}})) \tag{5}$$

where $A \in \mathbb{R}^{1 \times 1 \times (2C + 2C^{\text{cls}} + C^{\text{part}} + 2) \times C}$ is a filter bank whose purpose is to reduce the stacked representation back to the original $C$ channels. This is a useful design as it maintains the same representation dimensionality regardless of the number of tasks added. However, due to the compression, it may perform less well.

## 4 Experiments

### 4.1 Implementation details and training

The image encoder $\phi_{\text{enc}}^{\text{img}}$ is initialized from the pre-trained VGG-M model using sections conv1 to conv5. If the input to the network is an RGB image $\mathbf{x}^{\text{img}} \in \mathbb{R}^{H^{\text{img}} \times W^{\text{img}} \times 3}$, then, due to downsampling, the spatial dimension $H \times W \times C$ of $\mathbf{r}^{\text{img}} = \phi_{\text{enc}}^{\text{img}}(\mathbf{x}^{\text{img}})$ are $H \approx H^{\text{img}}/16$ and $W \approx W^{\text{img}}/16$. The number of feature channels is $C = 512$. As noted above, the decoders contain respectively subnetworks $\underline{\psi_{\text{dec}}}^{\text{cls}}$, $\underline{\psi_{\text{dec}}}^{\text{det}}$, and $\underline{\psi_{\text{dec}}}^{\text{part}}$ comprising layers fc6 and fc7 from VGG-M, followed by a randomly-initialized linear predictor with output dimension equal to, respectively, $C^{\text{cls}}$, $C^{\text{cls}} + 1$, and $C^{\text{part}} + 1$. Max pooling in SPP is performed in a grid of $6 \times 6$ spatial bins as in [14, 11]. The task encoders $\phi_{\text{enc}}^{\text{cls}}, \phi_{\text{enc}}^{\text{det}}, \phi_{\text{enc}}^{\text{part}}$ are given in sect. 2 and contain no parameter.

For training, each task is associated with a corresponding loss function. For the classification task, the objective is to minimize the sum of negative posterior log-probabilities of whether the image contains a certain object type or not (this allows different objects to be present in a single image). Combined with the fact that the classification branch uses sigmoid, this is the same as binary logistic regression. For the object and part detection tasks, decoders are optimized to classify the target regions as one of the $C^{\text{cls}}$ or $C^{\text{part}}$ classes or background (unlike image-level labels, classes in region-level labels are mutually exclusive). Furthermore, we also train a branch performing bounding box refinement to improve the fit of the selective search region as proposed by [11].

The fully connected layers used for softmax classification and bounding-box regression in object and part detection tasks are initialized from zero-mean Gaussian distributions with 0.01 and 0.001 standard deviations respectively. The fully connected layers used for object classification task and the adaptation layer $A$ (see eq. 5) are initialized with zero-mean Gaussian with 0.01 standard deviation.

All layers use a learning rate of 1 for filters and 2 for biases. We used SGD to optimize the parameters with a learning rate of $0.001$ for 6 epochs and lower it to $0.0001$ for another 6 epochs. We observe that running two iterations of recursion is sufficient to reach $99\%$ of the performance, although marginal gains are possible with more. We use the publicly available CNN toolbox MatConvNet [27] in our experiments.

## 4.2 Results

In this section, we describe and discuss experimental results of our models in two benchmarks.

**PASCAL VOC 2010 [10] and Parts [7]:** The dataset contains 4998 training and 5105 validation images for 20 object categories and ground truth bounding box annotations for target categories. We use the PASCAL-Part dataset [7] to obtain bounding box annotations of object parts which consists of 193 annotated part categories such as aeroplane engine, bicycle back-wheel, bird left-wing, person right-upper-leg. After removing annotations that are smaller than 20 pixels on one side and the categories with less than 50 training samples, the number of part categories reduces to $152$. The dataset provides annotations for only training and validation splits, thus we train our models in the train split and report results in the validation split for all the tasks. We follow the standard PASCAL VOC evaluation and report average precision (AP) and AP at $50\%$ intersection-over-union (IoU) of the detected boxes with the ground ones for object classification and detection respectively. For the part detection, we follow [7] and report AP at a more relaxed $40\%$ IoU threshold. The results for the tasks are reported in tab. 1.

In order to establish the first baseline, we train an independent network for each task. Each network is initialized with the VGG-M model, the last classification and regression layers are initialized with random noise and all the layers are fine-tuned for the respective task. For object and part detection, we use our implementation of Fast-RCNN [11]. Note that, for consistency between the baselines and our method, minimum dimension of each image is scaled to be $600$ pixels for all the tasks including object classification. An SPP layer is employed to scale the feature map into $6 \times 6$ dimensionality.

For the second baseline, we train a multi-task network that shares the convolutional layers across the tasks (this setting is called ordinary multi-task prediction in sect. 2.1). We observe in tab. 1 that the multi-task model performs comparable or better than the independent networks, while being more efficient due to the shared convolutional computations. Since the training images are the same in all cases, this shows that just combining multiple labels together improves efficiency and in some cases even performance.

Finally we test the full multinet model for two settings defined as update rules (1) and (2) corresponding to eq. 4 and 5 respectively. We first see that both models outperforms the independent networks and multi-task network as well. This is remarkable because our model consists of smaller number of parameters than the sum of three independent networks and yet our best model (update 1) consistently outperforms them by roughly 1.5 points in mean AP. Furthermore, multinet improves over the ordinary multi-task prediction by exploiting the correlations in the solutions of the individual tasks. In addition, we observe that update (1) performs better than update (2) that constraints the shared representation space to 512 dimensions regardless of the number of tasks, as it can be expected due to the larger capacity. Nevertheless, even with the bottleneck we observe improvements compared to ordinary multi-task prediction.

We also run a test case to verify whether multinet learns to mix information extracted by the various tasks as presumed. To do so, we exploit the predictions performed by these task in will be able to improve more with ground truth labels during test time. At test time we ground the classification label $\mathbf{r}^{\text{cls}}$ in the first iteration of multinet to the ground truth class labels and we read the predictions after one iteration. The performances expectedly in the three tasks improve to $90.1$, $58.9$ and $39.2$ respectively. This shows that, the feedback on the class information has a strong effect on class prediction itself, and a more modest but nevertheless significant effect on the other tasks as well.

**PASCAL VOC 2007 [10]:** The dataset consists of 2501 training, 2510 validation, and 5011 test images containing bounding box annotations for 20 object categories. There is no part annotations available for this dataset, thus, we exclude the part detection task and run the same baselines and our best model for object classification and detection. The results are reported for the test split and depicted in tab. 2. Note that our RCNN for the individual networks obtains the same detection score

| Method / Task | classification | object-detection | part-detection |
|---|---|---|---|
| Independent | 76.4 | 55.5 | 37.3 |
| Multi-task | 76.2 | 57.1 | 37.2 |
| Ours | **77.4** | **57.5** | **38.8** |
| Ours (with bottleneck) | 76.8 | 57.3 | 38.5 |

Table 1: Object classification, detection and part detection results in the PASCAL VOC 2010 validation split.

| Method / Task | classification | object-detection |
|---|---|---|
| Independent | 78.7 | 59.2 |
| MTL | 78.9 | 60.4 |
| Ours | **79.8** | **61.3** |

Table 2: Object classification and detection results in the PASCAL VOC 2007 test split.

in [11]. In parallel to the former results, our method consistently outperforms both the baselines in classification and detection tasks.

## 5    Conclusions

In this paper, we have presented multinet, a recurrent neural network architecture to solve multiple perceptual tasks in an efficient and coordinated manner. In addition to feature and parameter sharing, which is common to most multi-task learning methods, multinet combines the output of the different tasks by updating a shared representation iteratively.

Our results are encouraging. First, we have shown that such architectures can successfully integrate multiple tasks by sharing a large subset of the data representation while matching or even outperforming specialised network. Second, we have shown that the iterative update of a common representation is an effective method for sharing information between different tasks which further improve performance.

**Acknowledgments**

This work acknowledges the support of the ERC Starting Grant Integrated and Detailed Image Understanding (EP/L024683/1).

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
