[Reviews · NeurIPS 2016]

Reviewer 1

Summary

The paper presents a novel recurrent NN architecture, multinet, which enables integrated learning+inference in a multi-task setting. Each task's final response is obtained after several iterations of information transfer between all tasks. For instance the output of a classification task can be computed jointly with the output of object and parts detectors, each of which influences the output of other tasks.

Qualitative Assessment

I like the idea presented in the paper and as far as I am aware it is novel and relevant to NIPS. My two main issues are: 1. Generality: the approach is described in general but applied only to the vision domain. It would be great if the authors can add some examples and discussion (even only at the introduction) of other multi-task settings where they believe this approach is relevant. 2. Number of iterations: it is not clear to me from the text how many iterations are actually done? How was this number chosen? Are you looking for a convergence in the output space? Is the approach sensitive to the number of iterations? Minor issues: 1. Comparison to multi-task learning. Multi-task learning is about how a task can benefit from jointly learning. Multinet architecture enables multi-task learning but it also does more by considering multi-task relations during inference. I think this is worth while stressing in the introduction and in the comparison to "regular" multi-task learning. 2. I like the idea very much, but I find the experimental results are not too impressive, practically I am not sure I am convinced by the results to implement this method. 3. I find the discussion in introduction comparing simultaneous vs. sequential transfer confusing, it seems to indicate that the paper is doing sequential, starting from classifying the entire object the locating it and then locating it's parts. But isn't it actually done simultaneously in suggested multinet architecture? where the iterations are over all task jointly? 4. Typos: - line 92: "denoted by the symbol by" - line 92-93: "of the network (e.g. an image) of the network" - line 123: "in this cases" - line 279: "by the various task in will be able to"

Confidence in this Review

2-Confident (read it all; understood it all reasonably well)


Reviewer 2

Summary

This paper proposes multinet, a new integrated network in which not only deep image features are shared between tasks (representation sharing), but where tasks can interact in a recurrent manner by encoding the results of their analysis in a common shared representation of the data (output correlation). The proposed deep models are novel and seem technically sound, but the experimental evaluation may leave some room for improvement.

Qualitative Assessment

This paper is crystal clear and the main points are easily accessible. The key idea of integrated learning of representation sharing and output correlation is sound and well executed in the new architecture comprising CNNs, R-CNNs, RNNs and autoencoders. My main concern is regarding the experimental evaluation. There is clear room for improvement: (1) the authors are encouraged to use the standard VOC 2012 dataset instead of the more obsolete VOC 2010/2007 datasets--this makes direct comparison of different methods possible; (2) the baseline methods (Independent and Multi-task in Table 1) are too simple to justify the effectiveness of the proposed method, and more recent work on multi-task deep learning should be compared. Note that, although this paper contrasts itself clearly from the literature, it does not mean that it is enough to evaluate the proposed method only against simple baselines. The heterogeneous multi-task learning scenario is exactly the same with the state of the art work that won the MS COCO competition [8]. The multi-task network cascade (MNC) [8] is a sequential model that exploits the perception behavior of humans, i.e. from coarse-grained perception to fine-grained perception. I think MNC [8] is the most closely related method deserving empirical comparison (although MNC is sequential while the method of this paper is parallel). Another baseline method could be MT-CNN [31], which addresses the different convergence rates of different networks in the multi-task learning setting. Extending the heterogeneous multi-task learning scenario [31] to the classification, localization, and part detection as this work is straight-forward (MT-CNN is also a parallel learning method as this work). The final concern is on the model trainability. As multinet is a complex model comprising CNNs, R-CNNs, RNNs and autoencoders, I thus worry about the high difficulty in training the hybrid architecture. In particular, the autoencoders seem rather difficult: the back-propagation process (encoder) resorts to broadcast the outputs to the previous layers, which is quite ad hoc and requires more theoretical and empirical justification. I would expect to see what the authors can clarify in the rebuttal. -- Update after rebuttal: The authors have provided some responses on why the relevant baselines I mentioned were not empirically evaluated. I think the responses are reasonable and thus keep my original rating unchanged.

Confidence in this Review

2-Confident (read it all; understood it all reasonably well)


Reviewer 3

Summary

The paper proposes an approach for learning a shared latent representation across multiple tasks, using a recurrent neural network architecture. The proposed architecture uniformly models inputs as well as desired outputs as label spaces; the connection between the label spaces and the latent representation is made by means of encoders, decoders, and, notably, integrator functions that capture recurrent dependencies among different (timed) instantations of the latent representation. In that way, and different to traditional multi-task learning [1,5,25], the proposed architecture can leverage dependencies between outputs. The paper instantiates the proposed architecture in the vision domain, modeling jointly object classification, object detection, and part detection, using PSACAL VOC 2007 and 2010 data sets. The given experimental results indicate that the proposed architecture indeed outperforms traditional multi-task learning by small margins (in the 1-1.5 percent point range).

Qualitative Assessment

Positive points =============== + The paper is well written and follows a clear line of argumentation. + The proposed recurrent neural network architecture seems novel, technically correct, and is demonstrated to lead to performance improvements compared to a reasonable multi-task learning baseline. + The experiments also include results for a 'bottleneck' architecture that keeps the dimensionality of the latent representation fixed and demonstrates improved performance also in this case (albeit a little less than for the non-bottleneck version). Negative points =============== - The performance gain achieved by the proposed method is rather small. - Fig. 2 would benefit from instantiating more than just a single 'alpha' in addition to 'alpha = 0' in order to highlight the dependency among multiple tasks in addition to the dependencies across timesteps. - line 117: concatenated functions should be reversed - 144: 'input' (instead of 'output')

Confidence in this Review

2-Confident (read it all; understood it all reasonably well)


Reviewer 4

Summary

The authors propose a new architecture (multiunet) that allows learning multiple tasks in an integrated manner by recurrently updating a shared representation. For this architecture the authors provide to instantiations, both for a combination of three computer vision tasks: object classification, object detection and part detection. They are then evaluated on PASCAL VOC 2010 and 2007.

Qualitative Assessment

The content of the paper is well presented and the proposed integrated recurrent architecture seems very appealing. The considered set up with three computer vision tasks is very interesting and natural, and, to my knowledge, it was not explored before. However, the proposed instantiations of the network, given by the integrator functions (4) and (5) and the corresponding experimental results are not quite convincing. First, they show that sharing a representation leads to performance comparable to individual networks. However, it would be surprising if the performance would degrade - if the representation is good enough for detecting objects, it should be sufficient for classification as well. On the other hand, if I’ve understood correctly, the classification parts of both the multi-task and the recurrent networks have access to much richer training information - they are given the bounding boxes, while the independent classification network only knows the labels. Therefore improvement by 1 point seems lower than expected. In addition it is rather surprising that the multi-task network performs worse than the individual classification network, while being seemingly better on the detection task. Though, it is hard to estimate the significance of the difference. I could not find information on how many recurrent iterations were performed in the experiments (denoted by parameter t). It would be instructive to see how the performance of the system depends on this value.

Confidence in this Review

2-Confident (read it all; understood it all reasonably well)


Reviewer 5

Summary

This paper describes a recurrent neural network method for multi-task learning, which integrates knowledge between different labeling tasks (bounding box, parts, etc.) via a shared representation. The approach is applied to PASCAL VOC 2007 and 2010 data sets with favorable results.

Qualitative Assessment

Overall, I was very excited by this line of work, especially the idea of using recurrent neural networks for multi-task learning with iterative updates. The approach seems solid. However, the current paper is missing key comparisons to existing work both in the discussion and evaluation, including work that is directly relevant to their problem. Specifically, the authors should compare to Judy Hoffman's work in CVPR 2015 which addressed a similar problem of reusing knowledge between bounding box and parts detection tasks. The current paper does go beyond it with recurrent NNs, but this other work needs to be acknowledged and (ideally) compared to empirically. In addition, knowledge reuse via a shared representation is a key aspect of current work in lifelong learning and online multi-task learning (e.g., ICML'13, ICML'14, IJCAI'15) -- this work should be at least mentioned in the discussion. I also found it a bit unclear as to exactly what a task was in this work (paragraphs beginning at line 89). I believe that it is images from a single object category + their corresponding labels of a particular type (e.g., classification, bounding box, parts, etc.). Line 265 would seem to imply that transfer is only occurring between the different types of labels for a single object category. Is that correct? Are you exploring transfer / MTL across the different object categories as well? In addition to comparing to existing work empirically, the results should also report the standard deviation or standard error to assess significance. Are the results in Tables 1 and 2 averaged over all object categories? The clarity of the paper is excellent, and it is well-written overall.

Confidence in this Review

3-Expert (read the paper in detail, know the area, quite certain of my opinion)


Reviewer 6

Summary

The paper presents an integrated way of modelling multi-task problems where predictive outputs belong to different spaces (e.g. classification, segmentation, regression, etc). The proposed method relies on building a universal shared representation for all the predictive task. Although the paper focuses on an instantiation of the proposed framework using neural networks, it can be applied to other models, as long as we are able to define encoder-decoder functions.

Qualitative Assessment

The core idea of the proposed paper is very interesting. Indeed, I believe this idea illustrates in a very high level way, how humans work. Namely, we take advantage of related capabilities like segmenting, listening, etc when trying to visually identify a cat in an image. From a practical point of view, this might be understood as having an internal shared representation for each concept. This idea has been explored previously in other multi-modal works on the literature (Ngiam et al. "Multimodal Deep Learning"). Instead of analyzing this capability in a unsupervised manner (autoencoders), this paper addresses this idea from the heterogeneous predictive-task point of view. While the first question addressed by the authors regarding the universal representation can be understood as a multitask learning problem. The second question regarding different decision granularity raises questions from a transfer learning point of view. For example, is it possible to adapt a segmentation model into a classifier? All the previous comments were handled by the authors. I would suggest acceptance of the paper. I emphasize that the authors should correct their conclusions in lines 278-284 in the final version of the paper. My previous comment: "Section 4.2, lines 278-284: The results obtained by this strategy are not necessarily true. By grounding the classification label at the first iteration you are incurring in data leakage and, thereby, it is intuitive that the largest gain is obtained in the classification task. The important result on this experiment is the gain on the other tasks, which demonstrates the advantages of the shared representation. Thereby, the correct way of presenting this experiment requires to ground task T and verifying the gains on the other tasks T'. This would be analogous to having partial feedback from the user. For example, including the bounding box of the object for classification or including the final class to improve the part detection. Please consider to run these experiments as suggested or redefine your conclusions to be more accurate."

Confidence in this Review

2-Confident (read it all; understood it all reasonably well)